# Proof-of-Concept Analysis of B Cell Receptor Repertoire in COVID-19 Patients Undergoing ECMO by Single-Cell V(D)J and Gene Expression Sequencing

Alessia Gallo [1],[*],[†] , Nicola Cuscino [1],[†] , Claudia Carcione [2] , Rosalia Busà [1] , Pier Giulio Conaldi [1]
and Matteo Bulati [1],[*]

[1] Department of Research, IRCCS ISMETT (Istituto Mediterraneo per i Trapianti e Terapie ad Alta Specializzazione), 90127 Palermo, Italy
[2] Fondazione Ri.MED, Department of Research IRCCS ISMETT, Via Ernesto Tricomi 5, 90145 Palermo, Italy
[*] Correspondence: agallo@ismett.edu (A.G.); mbulati@ismett.edu (M.B.); Tel.: +39-091-21-92-649 (A.G. & M.B.)
[†] These authors contributed equally to this work.

**Abstract:** SARS-CoV-2, which causes COVID-19, has altered human activities all over the world and has become a global hazard to public health. Despite considerable advancements in pandemic containment techniques, in which vaccination played a key role, COVID-19 remains a global threat, particularly for frail patients and unvaccinated individuals, who may be more susceptible to developing ARDS. Several studies reported that patients with COVID-19-related ARDS who were treated with ECMO had a similar survival rate to those with COVID-19-unrelated ARDS. In order to shed light on the potential mechanisms underlying the COVID-19 infection, we conducted this proof-of-concept study using single-cell V(D)J and gene expression sequencing of B cells to examine the dynamic changes in the transcriptomic BCR repertoire present in patients with COVID-19 at various stages. We compared a recovered and a deceased COVID-19 patient supported by ECMO with one COVID-19-recovered patient who did not receive ECMO treatment and one healthy subject who had never been infected previously. Our analysis revealed a downregulation of FXYD, HLA-DRB1, and RPS20 in memory B cells; MTATP8 and HLA-DQA1 in naïve cells; RPS4Y1 in activated B cells; and IGHV3-73 in plasma cells in COVID-19 patients. We further described an increased ratio of IgA + IgG to IgD + IgM, suggestive of an intensive memory antibody response, in the COVID ECMO D patient. Finally, we assessed a V(D)J rearrangement of heavy chain IgHV3, IGHJ4, and IGHD3/IGHD2 families in COVID-19 patients regardless of the severity of the disease.

**Keywords:** COVID-19; BCR repertoire; ECMO; V(D)J; transcriptome

## 1. Introduction

Since December 2019, the severe acute respiratory syndrome coronavirus 2 (SARS-CoV-2) has posed a hazard to global public health. The coronavirus disease 2019 (COVID-19) shows a wide range of clinical manifestations, ranging from asymptomatic presentation to critical illness with severe pneumonia, acute respiratory distress syndrome (ARDS), or multiple organ failure [1]. The similarities between the worst SARS-CoV-2 consequences and seasonal influenza problems, such as ARDS or multiple organ failure, have suggested a role for extracorporeal membrane oxygenation (ECMO) implantation in patients with the most severe pulmonary decompensation [2,3]. It has recently been established that ECMO can be utilized as a rescue therapy due to the temporary replacement of lung and/or cardiac function [4]. Several studies have reported that patients with COVID-19-related ARDS who were treated with ECMO showed a survival rate comparable with those with COVID-19-unrelated ARDS [5–7]. Clearing the SARS-CoV-2 infection and hence influencing patients' clinical outcomes is also mediated by humoral and adaptive immune responses [8]. A crucial role is played by the B cell antigen receptor

(BCR) responsible for the recognition of pathogens. The processes of recombination and assembly of the variable and constant regions of the V, D, and J segments are crucial in generating an immense repertoire of antibodies responsible for the recognition of diverse pathogens [9]. Briefly, the antigen-binding domain of immunoglobulins is composed of two polypeptide chains. The exons that encode the antigen-binding domains are assembled from V (variable), D (diversity), and J (joining) gene segments by a process defined as "cut-and-paste" DNA rearrangements. This process, named V(D)J recombination, selects a pair of segments, introduces double-strand breaks next to each segment, deletes the intermediate DNA, and ligates the segments together. Rearrangements take place in a well-ordered way, with D-to-J joining proceeding before a V segment is joined to the rearranged D–J segments (Figure 1) [10]. Meeting a pathogen then unleashes rapid hypermutation (SHM) and class-switch recombination (CSR), thereby increasing antigen binding [11].

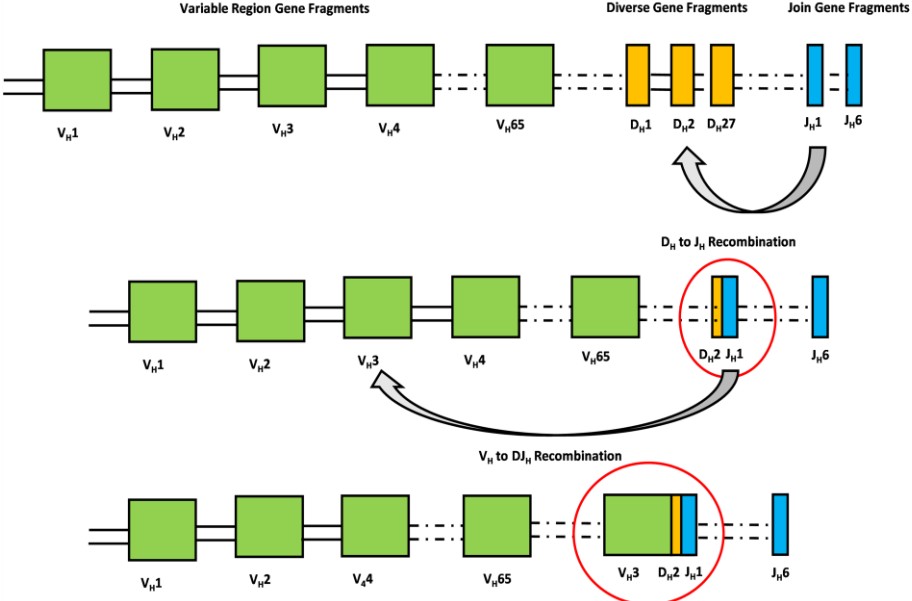

**Figure 1.** The antigen-binding domain of immunoglobulins is composed of two polypeptide chains, namely heavy and light chains. The exons that encode the antigen-binding domains of the heavy chain are assembled from V (variable), D (diversity), and J (joining) gene segments by a process defined as "cut-and-paste" DNA rearrangements. This process, named V(D)J recombination, selects a pair of segments, introduces double-strand breaks next to each segment, deletes the intermediate DNA, and ligates the segments together. Rearrangements take place in a well-ordered way, with D-to-J joining proceeding before a V segment is joined to the rearranged D–J segments. The rearrangement of the light chain is identical except for the absence of D gene fragments.

In this study, we used a single-cell approach to look at the dynamic changes in the transcriptomic BCR repertoire in patients with COVID-19 at various stages, comparing a recovered and a deceased COVID-19 patient who had been supported with ECMO with one COVID-19-recovered patient who had not received ECMO treatment and one healthy subject who had never previously been infected nor tested positive for SARS-CoV-2 antibodies.

## 2. Materials and Methods

### 2.1. Samples

Four subjects were enrolled in this study at IRCCS ISMETT: one COVID-19 patient on ECMO (survived), one COVID-19 patient on ECMO (died), one COVID-19 patient without ECMO (recovered), and one healthy control who tested negative for both a nasopharyngeal swab (NPS) and anti-Spike and anti-N IgG/IgM. The study was approved by the IRCCS ISMETT Institutional Research Review Board (IRRB 00/21) and the Ethics Committee of

IRCCS ISMETT, with all enrolled individuals signing a written informed consent form. Fresh blood samples were acquired from patients on day three of their hospitalization.

### 2.2. Preparation of Single-Cell Suspensions

Venous blood was collected in K3EDTA tubes (Greiner Bio-One GmbH, Kremsmunster, Austria). Peripheral blood mononuclear cells (PBMCs) were isolated by density gradient centrifugation on Lympholyte Cell Separation Media (Cedarlane Laboratories Limited, Burlington, ON, Canada). Afterwards, CD19+ B cells were separated from PBMCs by immuno-magnetic sorting using anti-CD19 magnetic microbeads (REAlease CD19 MicroBeads Kit, Miltenyi Biotec, Auburn, CA, USA). The CD19+ B cells obtained from immuno-magnetic sorting displayed a purity yield greater than 98%, which was determined by flow cytometry analysis.

### 2.3. Droplet-Based Single-Cell Sequencing

Using the single-cell 3′ Library and Gel Bead Kit V3.1 (10X Genomics, 1,000,121) and the Chromium Single Cell G Chip Kit (10X Genomics, 1,000,120), the cell suspension was loaded onto the Chromium Single Cell Controller (10X Genomics) to generate single-cell gel beads in an emulsion according to the manufacturer's protocol. In short, single cells were suspended in phosphate-buffered saline containing 0.04% bovine serum albumin. Approximately 1200 cells/µL were added to each channel, and the target number of cells to be recovered was estimated to be approximately 6000. Captured cells were lysed, with the released RNA barcoded through reverse transcription in individual GEMs. Reverse transcription was performed on a Veriti 96 Well Thermal Cycler (ThermoFisher) at 53 °C for 45 min, followed by 85 °C for 5 min, and then held constant at 4 °C. The generated cDNA was then amplified, with the quality assessed using the 4200 TapeStation System (Agilent). According to the manufacturer's instructions, scRNA-seq libraries were constructed using the Single Cell 3′ Library and Gel Bead Kit V3.1. Finally, the libraries were sequenced using an Illumina NextSeq500/550 High Output Reagent Cartridge v2 300 cycle sequencer, with a sequencing depth of at least 100,000 reads per cell using the paired-end strategy.

### 2.4. Single-Cell RNA Sequencing

First, Cell Ranger v 6.0.0 (10X Genomics) was used to demultiplex the cellular barcodes and align the reads to the human transcriptome (human reference version GRCh38) for each sample [12]. Second, each output, which was a raw unique molecular identifier (UMI) count matrix, was transformed into a Loupe Browser object. We filtered out genes that were expressed in less than five cells by using the Loupe Browser v 6.0.0 function. Several criteria were then applied to each dataset to remove cells of low quality: cells with fewer than 200 genes or more than 6000 genes. Data processing and analysis were performed with Prism GraphPad V5.0d software (GraphPad Software, San Diego, CA, USA).

### 2.5. Single-Cell V(D)J Sequencing and Data Processing

Single-cell V(D)J sequencing was performed following the protocol provided by the 10X Genomics Chromium Single Cell Immune Profiling Solution. The analysis pipelines in Cell Ranger (10X Genomics, version 6.0.0) were used for single-cell sequencing data processing and were loaded in the Loupe V(D)J Browser v 4.0.0. V(D)J sequence assembly and paired clonotype calling were performed using cellranger vdj with reference = refdata-cellranger-vdj-GRCh38-alts-ensembl-6.0 for each sample. Data analysis was performed using the tidyverse package v1.3.0.

## 3. Results

### 3.1. Study Design and Profiling of B Cells

In order to evaluate the different signatures of B cell receptors in the B lymphocytes of COVID-19 patients that needed extracorporeal membrane oxygenation (ECMO) in response

to acute respiratory distress syndrome (ARDS), we performed scRNA-seq and single-cell BCR sequencing on the CD19+ immune cells of a COVID-19 patient under ECMO who later survived the treatment (COVID ECMO S), a COVID-19 patient under ECMO who later died (COVID ECMO D), one COVID-19-recovered patient who did not receive ECMO support (COVID R), and one healthy control who tested negative for both a nasopharyngeal swab (NPS) and anti-Spike and anti-N IgG/IgM. In Table 1, we report the characteristics of the patients included in this study. After filtering, 7884 B cells were obtained for the single-cell transcriptome data and 7733 B cells were obtained for V(D)J analysis. The scRNA-seq and single-cell paired BCR analysis were then combined, with the study limited to 5830 B lymphocytes having full-length productive paired IGH-IGK/IGL.

**Table 1.** Characteristics of patients and control.

| ID Patients | Age | Gender | Comorbidities | Severity | Clinical Profile | Co-Infections | Treatments | Hospitalization (Days) |
|---|---|---|---|---|---|---|---|---|
| COVID ECMO S | 74 | M | Vasculopathy and Diabetes | C | ARDS | - | Dexamethaso | 30 total (10 ECMO) |
| COVID ECMO D | 67 | M | HTN | C | ARDS and Sepsis | MDR *Xanthomonas* (VAE), MDR Acinetobacter (UTI), C.albicans (BSI) | CPAP and Dexamethasone | 40 total (12 NIV + 7 ECMO) |
| COVID R | 42 | F | - | A | - | - | - | |
| HEALTHY CTRL | 45 | M | - | - | - | - | - | |

Abbreviations: HTN, hypertension; C, critical; A, asymptomatic; ARDS, acute respiratory distress syndrome; MDR, multidrug-resistant; VAE, ventilator-associated events; UTI, urinary tract infection; BSI, bloodstream infection; CPAP, continuous positive airways pressure; NIV, non-invasive ventilation; ECMO, extra corporeal membrane oxygenation.

### 3.2. Features of B Cell Subsets

We started our analysis by assigning cell identities based on B cell surface marker expression as indicated below. According to the average log fold change of the canonical markers, we identified four clusters: (1) memory B cells; (2) naïve cells; (3) activated B cells, and (4) plasma cells (Figure 2A). Memory B cell subsets were identified by the presence of the typical memory marker CD27, while naïve B cells expressed the heavy chain IgD immunoglobulin (IGHD) [13]. Activated B cells expressed high levels of CD79, a typical marker of B cell activation [14], while plasma cells, together with CD79, expressed a high level of X-box binding protein 1 (XBP1), a transcriptional regulator critical for supporting the cellular reprograming activities during B-to-plasma-cell transition, which permits antibody release during terminal differentiation [15]. All B cell subpopulations analyzed expressed the membrane-spanning 4A1 (MS4A1 or CD20) marker, which unequivocally identifies B cells. In order to assess differences in the B cell populations present in the four samples, we investigated the relative frequencies of the four clusters in each sample (Figure 2B). In general, naïve cells and activated B cells accounted for the largest portion of B cells in each sample. Among COVID-19 patients, it is notable that, in the COVID ECMO D patient, the naïve cell to activated B cell ratio was different from the other samples with a shift towards the activated B cell population (Figure 2B,C). We also investigated the transcriptional signatures of the COVID-19 patients' B cell populations (Figure 2D).

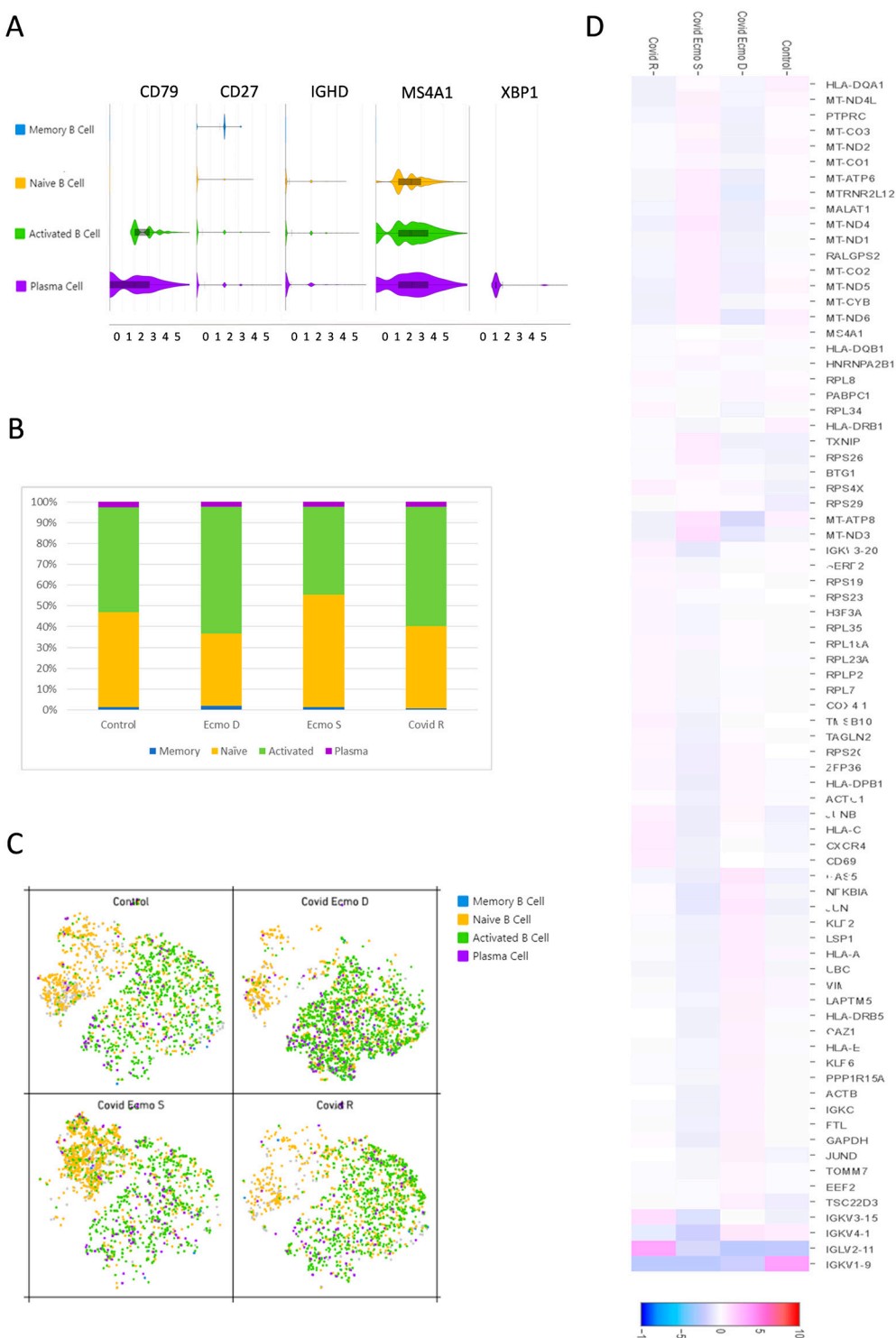

**Figure 2.** Features of B cell subsets. (**A**) Violin plots show the expression distribution of canonical cell markers in B cell subsets. (**B**) Proportion of B cell clusters in each sample. (**C**) t-SNE projection of B cells from all four samples. Each dot corresponds to a single cell, colored according to cell clusters. (**D**) Heat map representation of differentially expressed genes of B cell populations in the COVID ECMO S, COVID ECMO D, COVID R, and the healthy control. Genes with a maximum adjusted *p*-value of 0.01 and an absolute value of log2 (fold change) >0.5 were considered to be differentially expressed genes.

### 3.3. Transcriptional Signatures of Different B Cell Subpopulations of COVID-19 Patients

Once the clusters present in the B cell populations of the COVID ECMO S, COVID ECMO D, COVID R, and healthy control were established, we investigated the transcriptional profiles of the clusters in depth to find clue genes and pathways potentially involved in the worsening of clinical outcomes (Figure 3A).

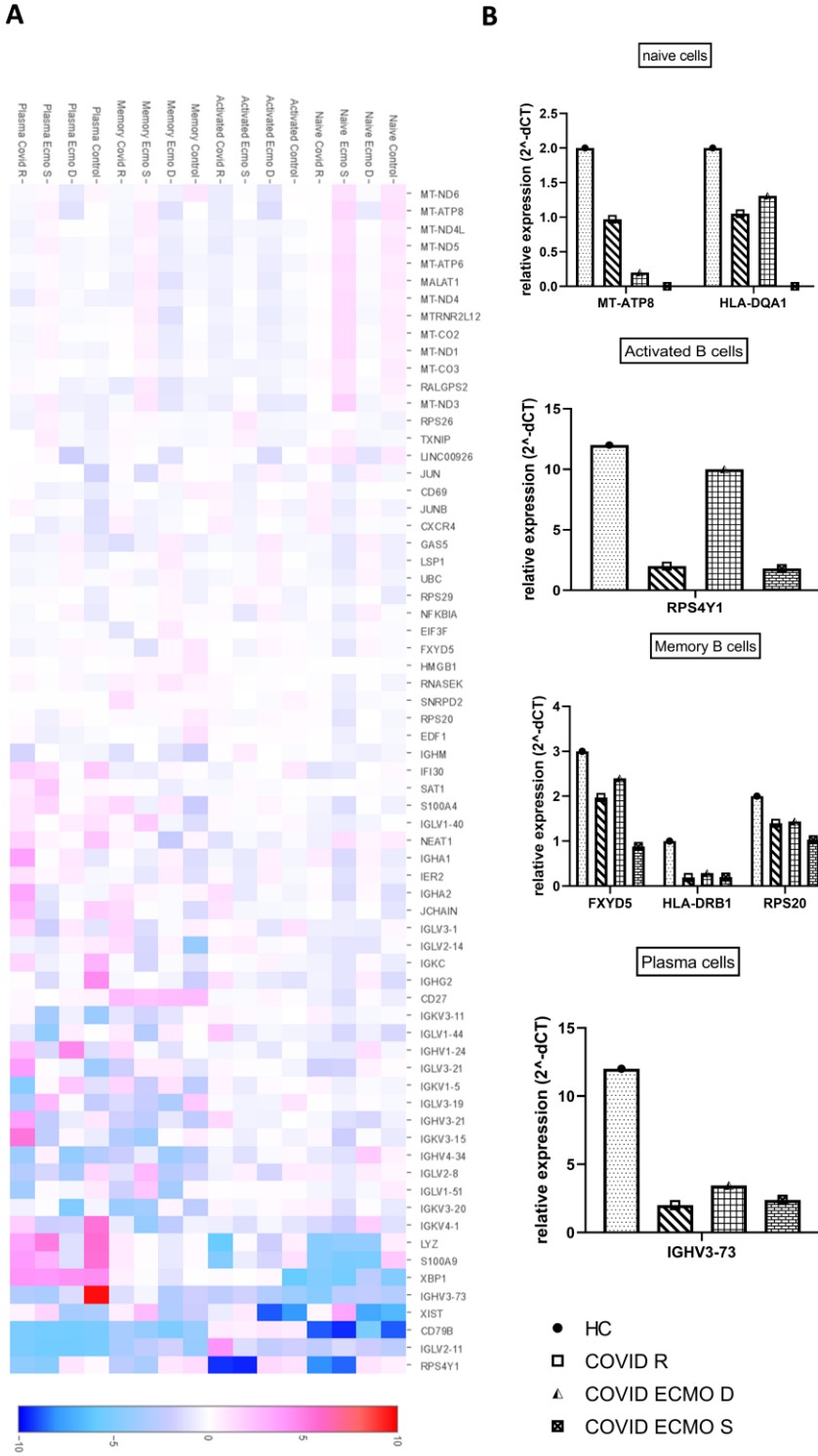

**Figure 3.** Features of B cell subpopulations. (**A**) Heat map representation of differentially expressed genes of B cell subpopulations (naïve cells, activated B cells, memory B cells, and plasma cells) of the COVID ECMO S, COVID ECMO D, COVID R, and healthy control. Genes with a maximum adjusted

*p*-value of 0.01 and an absolute value of log2 (fold change) >0.5 were considered to be differentially expressed genes. (**B**) The graphs show the sequencing results of highly deregulated mRNA in the COVID ECMO S, COVID ECMO D, COVID R, and healthy control.

Among the deregulated genes in the naïve B cells, we found a downregulation of the major histocompatibility complex, Class II, DQ Alpha 1 (HLA-DQA1), and the mitochondrial gene MT-ATP8 (Mitochondrially Encoded ATP Synthase Membrane Subunit 8) in COVID-19-infected subjects compared to the uninfected subject (Figure 3B). Interestingly, we discovered that MT-ATP8 expression levels were associated with disease severity; in particular, the two COVID-19 patients receiving ECMO had near-zero MT-ATP8 expression compared to the no-ECMO patient and the healthy control. The most notable downregulated gene in activated B cells was the ribosomal protein S4, Y-Linked 1 (RPS4Y1), which has been linked to HLA-DQA1 and has been implicated in resistance to treatment with corticosteroids and cyclosporin A (Figure 3B). The memory B cell cluster showed downregulation of three genes: FXYD, Domain-Containing Ion Transport Regulator 5; HLA-DRB1, a beta chain of antigen-presenting major histocompatibility complex class II (MHCII) molecule; and RPS20, ribosomal protein S20 (Figure 3B). The profiling of plasma cell clusters revealed a considerable downregulation of an immunoglobulin heavy chain variable region, IGHV3-73, which has been described as permitting antigen-binding activity and immunoglobulin receptor-binding activity (Figure 3B).

### 3.4. IgH Class Switching of B Cells and CDR3 Length and Specific Rearrangements of V(D)J Genes

To delineate the dynamic changes in IgH class switching, which is crucial for a comprehensive analysis of the BCR repertoire, we evaluated the distribution of Immunoglobulins, including IgA, IgD, IgG, and IgM in the COVID ECMO S, COVID ECMO D, COVID R, and healthy control (Figure 4A). In the healthy control, 61.3% of B cells expressed the IgM isotype, followed by IgG (22.4%), IgA (16.2%), and IgD (0.1%). Of COVID ECMO D patient B cells, 55.8% expressed the IgM isotype, followed by IgG (28.6%), IgA (18.2%), and IgD (2.3%). Of COVID ECMO S patient B cells, 70.2% expressed the IgM isotype, followed by IgG (18.5%), IgA (11.3%), and IgD (0.1%), while 65.2% of COVID R patient B cells expressed the IgM isotype, followed by IgG (13.9%), IgA (18%), and IgD (2.8%) (Figure 4B). Interestingly, compared to the other subjects involved in the study, the ratio of (IgA + IgG) to (IgD + IgM) increased in the COVID ECMO D patient, suggesting an intensive antibody response.

The distribution of clonally increased B cells in the COVID ECMO S, COVID ECMO D, COVID R, and healthy control was studied. The percentage of clonally expanded cells in COVID ECMO D patient B cells was 1.97%, which was somewhat higher than in the other three subjects, specifically: 0.56% in COVID ECMO S, 1.0206% in COVID R, and 1.14% in the healthy control (Figure 4C).

The CDR3 length of the BCR heavy chain in the COVID ECMO S, COVID ECMO D, and COVID R patients, and in the healthy control, ranged from 6 to 51 amino acids (aa), with an average of 14 aa for each sample (Figure 4D), and no significant difference between the samples.

In order to investigate the V(D)J rearrangements of the BCR heavy chain in our four samples, we detected the differences in usage frequency of the V, D, and J gene segments. A total of 53 IGHV gene segments, 27 IGHD gene segments, and 6 IGHJ gene segments were identified from all of the B cells. We generated a distribution histogram of IGHV, IGHD, and IGHJ gene usage frequency for the total number of B cells in our samples. We assessed that IGHJ4 was most frequently used in all samples, followed by IGHJ6, IGHJ5, and IGHJ3, but with no significant differences related to the disease status. IGHJ1 and IGHJ2 showed the lowest utilization (Figure S1).

The analysis of selective usage of IGHD and IGHV genes is summarized in Figures S2 and S3. In our samples, we detected an over-representation of the IGHD3 and IGHD2 families. Among the IGHD3 genes, the most expressed is IGHD3-22, which was highly

expressed in both COVID-19 patients receiving ECMO (22.94% COVID ECMO D; 17.51% COVID ECMO S), compared to COVID R (15.45%) and the healthy control (13.60%), while the least expressed is IGHD3-3, in both COVID patients that underwent ECMO (8.72% COVID ECMO D; 10.31% COVID ECMO S), compared to COVID R (17.89%) and the healthy control (11.48%) (Figure S2).

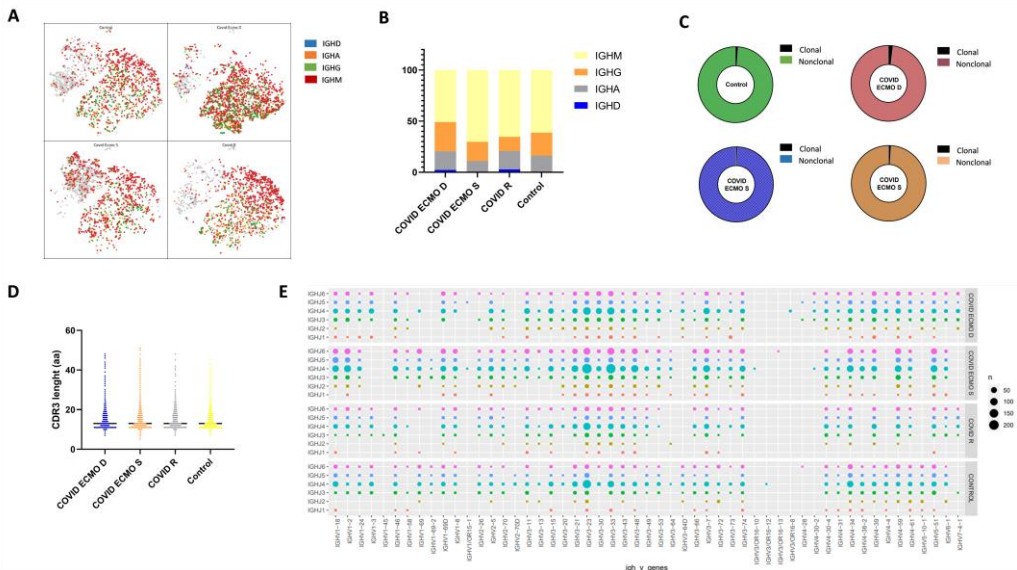

**Figure 4.** (**A**) t-SNE plot of B cells colored by immunoglobulin heavy chain expression. (**B**) The proportion of different immunoglobulin isotypes in each sample. (**C**) The pie plot shows the proportion of clonally expanded B cells in the COVID ECMO S, COVID ECMO D, and COVID R patients and in the healthy control. (**D**) The distribution of immunoglobulin heavy chain (IGH) CDR3 length. (**E**) The bubble chart shows the usage frequency of V–J gene combination in each sample.

Regarding the IGHV genes, we detected that the most represented is the IGHV3 family, especially IGHV3-23, which was expressed less in both COVID patients that underwent ECMO (8.85% COVID ECMO D; 10.14% COVID ECMO S), compared to COVID R (13.70%) and the healthy control (13.14%), and IGHV3-33, with no significant differences related to the disease status (Figure S3).

A total of 318 unique V–J combinations were identified in all of the whole B cells (Figure 4E). The top paired V–J frequency in the COVID patients receiving ECMO was IGHV3-33/IGHJ4 (4.68% COVID ECMO D; 5.33% COVID ECMO S), compared to COVID R (4.02%) and the healthy control (4.22%).

## 4. Discussion

In the present study, we aimed to perform a comprehensive analysis of the BCR repertoire, which is the genetic source of neutralizing antibodies (nAbs) [16], in COVID-19 patients that underwent ECMO using a single-cell approach. Since the quantification of gene expression is necessary to uncover disease etiology and progression [17], our goal was to shed light on the complex mechanisms that drive the immune response to the virus stimulus. The cohort included a COVID-19 patient who died after receiving ECMO (COVID ECMO D), a COVID-19 patient who survived the treatment (COVID ECMO S), a COVID-19-recovered patient without ECMO support (COVID R), and one healthy control who tested negative for both a nasopharyngeal swab (NPS) and anti-Spike and anti-N IgG/IgM. The first step was to identify four clusters—memory B cells, naïve cells, activated B cells, and plasma cells—on CD19+ immune B cells based on the average log fold change of the canonical markers, and we assessed the relative frequencies of the four clusters in each sample. We found that naïve cells and activated B cells accounted for the largest portion of B cells in each sample but, in the COVID ECMO D patient, the naïve cell to

activated B cell ratio was different from the other samples, with a shift towards activated B cell populations.

The transcriptomic analysis highlights different signatures of the B cell subpopulations among the subjects studied. In particular, the naïve cells of COVID-19-infected subjects showed downregulation of MT-ATP8 and HLA-DQA1. The first is a mitochondrial gene that encodes for an ATP synthase responsible for changing cellular energy capacity, increasing mitochondrial oxidative stress and/or modulating apoptosis [18]. Interestingly, we observed that its levels of expression were related to the severity of the disease. Indeed, the two COVID-19 patients receiving ECMO showed lower MT-ATP8 expression, close to zero, compared to the no-ECMO patient and the healthy control. The second is known to bind peptides derived from antigens that access the endocytic route of antigen-presenting cells (APC) and present them on the cell surface for recognition by the CD4 T-cells. In activated B cells, the ribosomal protein S4, Y-Linked 1 (RPS4Y1), described as being involved, together with HLA-DQA1, in a resistance to the treatment with corticosteroids combined with cyclosporin A, commonly used in autoimmune diseases, auto-inflammatory diseases, and transplant rejection [14], was significantly downregulated. The memory B cell cluster showed downregulation of three genes: (1) FXYD, Domain-Containing Ion Transport Regulator 5, a glycoprotein that functions in the upregulation of chemokine production and is involved in the reduction of cell adhesion via its ability to downregulate E-cadherin; (2) HLA-DRB1, a beta chain of the antigen-presenting major histocompatibility complex class II (MHCII) molecule, known to guide antigen-specific T-helper effector functions, both in antibody-mediated immune responses and macrophage activation, to ultimately eliminate infectious agents and transformed cells [19–25]; and (3) RPS20, ribosomal protein S20, for which downregulation contributes to a stress phenotype by suppressing genomic and cellular stability [26]. Plasma cell cluster profiling showed significant downregulation of an immunoglobulin heavy chain variable region, IGHV3-73, described as enabling antigen-binding activity and immunoglobulin receptor-binding activity and predicted to be involved in several processes, including activation of the immune response and the defense response to other organisms, as well as phagocytosis.

We also evaluated the clonality of B cells in our cohort and found no significant differences, apart from a slight difference in clonally expanded cells in COVID ECMO D patient B cells (1.97%), versus the B cells of the other three subjects studied: 0.56% in COVID ECMO S, 1.0% in COVID R, and 1.14% in the healthy control. We also evaluated the B cell immunoglobulin distribution (IgA, IgD, IgG, and IgM) and found that the ratio of IgA + IgG to IgD + IgM was greater in the COVID ECMO D patient, suggesting an intensive memory antibody response. Lastly, we assessed the V(D)J rearrangements of the BCR heavy chain of our four samples and determined that the IGHV3, IGHJ4, and IGHD3/IGHD2 families were the most frequently used in all samples that we analyzed. These results are consistent with previously published data, in which the authors also reported the highest frequency of the IGHV3/IGHJ4 pair in symptomatic patients [27]. In addition, among the IGHD3 genes, the most expressed was IGHD3-22, which was highly expressed in both COVID-19 patients that underwent ECMO (22.94% COVID ECMO D; 17.51% COVID ECMO S), compared to COVID R (15.45%) and the healthy control (13.60%). In a recent study [28], the authors identified more than 100 anti-SARS-CoV-2 antibodies containing a conserved YYDRxG motif exclusively encoded by the IGHD3-22 gene. The presence of this motif is strictly related to a high neutralizing activity against different SARS-CoV-2 variants, as well as other SARS-related coronaviruses [28]. Conversely, IGHD3-3 is expressed less in both COVID-19 patients that underwent ECMO (8.72% COVID ECMO D; 10.31% COVID ECMO S), compared to COVID R (17.89%) and the healthy control (11.48%). Finally, a total of 318 unique V–J combinations were identified from all the B cells. The top paired V–J frequency in the COVID-19 patients, COVID ECMO D and COVID ECMO S, were 4.68% and 5.33%, respectively, compared to COVID R (4.02%) and the healthy control (4.22%). Our study, through the integration of transcriptomic data and single-cell paired BCR profiles, revealed BCR repertoire changes in COVID-19 patients with severe illness,

which is consistent with other studies focusing on subjects infected and/or vaccinated against SARS-CoV-2 subjects [29].

## 5. Conclusions

The present study aimed to perform a comparative analysis of the BCR repertoire of COVID-19 patients with different clinical profiles and outcomes: a COVID-19 patient receiving ECMO who later died, a COVID-19 patient receiving ECMO who later survived treatment, a COVID-19 patient who recovered without ECMO support, and one healthy control. After the identification of four clusters on CD19+ immune B cells—memory B cells, naïve cells, activated B cells, and plasma cells—we evaluated the BCR repertoires from different points of view. (1) Firstly, we assessed the transcriptome highlighting different signatures of the B cell subpopulations in COVID-19-infected subjects showing a downregulation of FXYD, HLA-DRB1, and RPS20 in memory B cells; a downregulation of MT-ATP8 and HLA-DQA1 in naïve cells; a downregulation of RPS4Y1 in activated B cells; and a downregulation of IGHV3-73 in the plasma cells. (2) We then evaluated the clonality of B cells with no strong differences found. (3) We appraised the B cell immunoglobulin distribution, finding an increased ratio of IgA + IgG to IgD + IgM in the COVID ECMO D patient, suggesting an intensive memory antibody response. (4) Lastly, we assessed the V(D)J rearrangements of the BCR heavy chain in our four samples and found that the IGHV3, IGHJ4, and IGHD3/IGHD2 families were the most frequently used.

The study's main weakness is the small sample size, which prevents the results from being meaningful for stating any of the processes involved in the immune response to SARS-CoV-2. However, given the uniqueness of the disease's characteristics and the molecular approach used, single-cell sequencing, which is based on a large number of events/cells and analyzes each cell as an independent sample, we hope that our computational analysis will provide novel insights for large population cohort studies. Our findings uncover that the V(D)J rearrangements of the BCR heavy chain of ECMO COVID-19 patients analyzed are consistent with other studies focusing on subjects infected with and/or vaccinated against SARS-CoV-2. As a result, we plan to conduct future clinical studies on large cohorts of COVID-19 patients in order to better analyze and compare the BCR repertoire in patient subgroups.

**Supplementary Materials:** The following supporting information can be downloaded at: https://www.mdpi.com/article/10.3390/cimb45020095/s1, Figure S1: Selective usage of J gene segments in each sample; Figure S2: Selective usage of V gene segments in each sample; Figure S3: Selective usage of D gene segments in each sample.

**Author Contributions:** Conceptualization, A.G. and M.B.; methodology, A.G., C.C., M.B. and R.B.; software, N.C.; validation, C.C. and A.G.; formal analysis, N.C., A.G., R.B. and M.B.; investigation, A.G. and M.B.; writing—original draft preparation, M.B., R.B. and A.G.; writing—review and editing, A.G., M.B. and P.G.C.; visualization, M.B., R.B. and A.G.; supervision, A.G., M.B. and P.G.C. All authors have read and agreed to the published version of the manuscript.

**Funding:** This research was funded by Ministero della Salute from current research funds in 2022 (Ricerca Corrente 2022) and by EU funding within the MUR PNRR Extended Partnership initiative on Emerging Infectious Diseases (Project no. PE00000007, INF-ACT).

**Institutional Review Board Statement:** The study was conducted in accordance with the Declaration of Helsinki and was approved by the IRCCS ISMETT Institutional Research Review Board (IRRB 00/21) and the Ethics Committee of ISMETT, with all enrolled individuals signing the written informed consent form.

**Informed Consent Statement:** Informed consent was obtained from all subjects involved in the study.

**Data Availability Statement:** The raw data supporting the conclusions of this article will be made available by the authors, without undue reservation, to any qualified researcher.

**Acknowledgments:** We would like to thank Stefano Anzani for assistance in using R packages.

**Conflicts of Interest:** The authors declare no conflict of interest. The funders had no role in the design of the study; in the collection, analyses, or interpretation of data; in the writing of the manuscript; or in the decision to publish the results.

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
