# Peer review of "Proof-of-Concept Analysis of B Cell Receptor Repertoire in COVID-19 Patients Undergoing ECMO by Single-Cell V(D)J and Gene Expression Sequencing"

_cimb, doi:10.3390/cimb45020095_

Round 1
Reviewer 1 Report
REVIEWER'S REPORT
Manucsript title:
Proof of concept analysis of B cell receptor repertoire in COVID-2 19 patients under ECMO by single-cell V(D)J and gene expression sequencing. (Authors: Alessia Gallo, Nicola Cuscino, Claudia Carcione, Rosalia Busà, Pier Giulio Conaldi, Matteo Bulati).
This manuscript, in my opinion, can definitely be accepted, but only after certain text edits and supplementations. Despite the fact that the manuscript is written in acceptable English, I would nevertheless recommend editing the entire text as it may significantly improve the text quality.
Virtually the whole text in the Abstract must be more thoroughly stated; especially, the statement in lines 12-14 should be replaced by "... which causes coronavirus disease 2019 (COVID-19), has altered human activities all over the world and has become a global hazard to public health"; in lines 14-17 " Despite considerable advancements in pandemic containment techniques, in which vaccination played a key role, COVID-19 remains a global threat, particularly for frail patients and unvaccinated individuals who may be more susceptible to developing acute respiratory distress syndrome (ARDS)."; in line 19 "... had a similar survival rate to those with COVID-19-unrelated ARDS."; and in lines 19-25 " In order to shed light on the potential mechanisms underlying the COVID-19 infection, we conducted this proof of concept study using a single-cell V(D)J and gene expression sequencing of the B cell to examine the dynamic changes in transcriptomic BCR repertoire present in patients with COVID-19 at various stages. We compared a recovering and a deceased COVID-19 patient supported by ECMO with one COVID-19-recovered patient who did not receive ECMO treatment and one healthy subject who had never been infected previously."
In Introduction. Rewrite the sentence in lines 29-30 as follows: " Since December 2019, the severe acute respiratory syndrome coronavirus 2 has posed a hazard to global public health (SARS-CoV-2)."; in lines 33-38, the sentence might be replaced by " The similarities between the worst SARS-CoV-2 consequences and seasonal influenza problems, such as ARDS or multiple organ failure, suggested a role for extracorporeal membrane oxygenation (ECMO) implantation in patients with the most severe pulmonary de-compensation."; in lines 36-38, the sentence might be paraphrased as " It has recently been established that ECMO can be utilized as a rescue therapy due to the temporary replacement of lung and/or cardiac function."; the sentence in lines 40-42 should be replaced by " Clearing SARS-CoV-2 infection and hence influencing patient clinical outcomes is also mediated by humoral and adaptive immune responses."; in lines 55-59, the sentence could be replaced by " In this study, we used a single-cell approach to look at the dynamic changes in transcriptomic BCR repertoire in patients with COVID-19 at various stages, comparing a recovered and a deceased COVID-19 patient supported with ECMO with one COVID-19-recovered patient without ECMO treatment, and one healthy subject never infected before and tested for SARS-CoV-2 antibodies."
In Materials and Methods. In lines 62-65, the sentence should be rewritten as " Four subjects were enrolled in this study at the IRCCS ISMETT: one COVID-19 patient on ECMO (survived), one COVID-19 patient on ECMO (died), one COVID-19 recovered patient without the ECMO condition, and one Healthy control who tested negative for both nasopharyngeal swab (NPS) and anti-Spike and anti-N IgG/IgM."; in lines 68-69, the sentence might be replaced by " Fresh blood samples were acquired from patients on day three of their hospitalization."
In Results. Study Design and Profiling of B Cells. The lines 125-127 should be paraphrased as " The scRNA-seq and single-cell paired BCR analysis were then combined, and the study was limited to 5,830 B lymphocytes having full-length productive paired IGH-IGK/IGL."
In Transcriptional signatures of different B cell subpopulations of COVID-19 patients. Lines 177-180 should be paraphrased as follows " Interestingly, we discovered that MT-ATP8 expression levels were associated to disease severity; in particular, the two COVID-19 patients getting ECMO had near-zero MT-ATP8 expression compared to the no-ECMO patient and the Healthy control."
In Features of B Cell Subsets. Lines 177-185 should be rewritten as "Interestingly, we discovered that MT-ATP8 expression levels were associated to disease severity; in particular, the two COVID-19 patients getting ECMO had near-zero MT-ATP8 expression compared to the no-ECMO patient and the Healthy control. The most notable downregulated gene in activated B cells was ribosomal protein S4, Y-Linked 1 (RPS4Y1), which has been linked to HLA-DQA1 and has been implicated in the resistance to treatment with corticosteroids and cyclosporin A. (Figure 2B)."; in lines 186-188, the sentence might be replaced by " The profiling of plasma cells clusters revealed a considerable downregulation of an Immunoglobulin Heavy Chain Variable Region, IGHV3-73, which has been described to permit antigen binding activity and immunoglobulin receptor binding activity (Figure 2B)."; the sentence in lines203-207, should be replaced by "The distribution of clonally increased B cells in COVID ECMO S, COVID ECMO D, COVID R, and Healthy controls was studied. The percentage of clonally expanded cells in COVID ECMO D patient B cells was 1.97%, which was somewhat higher than in the other three subjects' B cells, specifically: 0.56% in COVID ECMO S, 1.0 206% in COVID R, and 1.14% in the Healthy control (Figure 3C)."; in lines 210-211 " ....for each sample (Figure 3D), with no significant difference between the samples."
In Discussion. In line 245, the first phrase should begin " In the present study, we aimed (or wanted) to..." (not study's goal but the authors' goal); the sentence in lines 248-253 should be written as " The cohort included a COVID-19 patient who died after receiving ECMO (COVID ECMO D), a COVID-19 patient who survived the treatment (COVID ECMO S), a COVID-19 recovered patient without ECMO support (COVID R), and one healthy control who tested negative for both nasopharyngeal swab (NPS) and anti-Spike and anti-N IgG/IgM. "; line 255 should be slightly revised to read "... B cells sequenced, based on the average log fold change of the canonical markers, and we assessed..."
The Discussion (lines 315-330) focuses mainly upon the description of experimental technique rather than the interpretation of the results, and reading the conclusions(lines 315-330) it appears that the objective of this work is the technique for doing it out. So, the question is what is the true meaning of this work?
The statement in lines 332-334 of Limitation of the study (page 10) should be changed by " It is important to highlight that the study's main weakness is the small sample size, which prevents the results from being meaningful for stating any of the processes involved in the immune response to SARS-CoV-2. "; I suggest changing the phrase in lines 334–338 to "However, given the uniqueness of the disease characteristics and the molecular approach used, single-cell sequencing, which is based on a large number of events/cells and analyzes each cell as an independent sample, we hope that our computational analysis will provide novel insights for large population cohort studies.";
Other notes:
I recommend to improve the quality (color sharpness) of Figures 1 D and Figure 2 A.
Finally, I recommend that the structure/composition or formal scheme of the antigen-binding domain of immunoglobulins be provided not only in the text (as described in lines 46-53), but also illustrated with detailed explanations, so that the reader may better understand the context of this article.
.
Author Response
Palermo 6th February 2023
Dear Editor,
Please find enclosed a revised version of the Case Report entitled “Proof of concept analysis of B cell receptor repertoire in COVID-19 patients under ECMO by single-cell V(D)J and gene expression sequencing.” ID: cimb-2196562 by Gallo et al.
As required, together with the revised manuscript, we now resubmit a point by point response to the reviewer’s criticisms (see below).
We thank the Referees for their helpful comments that greatly improved the manuscript. We have tried to do our best to respond to the points raised.
The Referees have brought up some good points and we appreciate the opportunity to clarify our research objectives and results. As indicated below, we have checked all the general and specific comments provided by the Referees and have made necessary changes accordingly to their indications.
Sincerely,
Alessia Gallo
Alessia Gallo, PhD
Department of Research,
IRCCS ISMETT, via Tricomi 5, 90127 Palermo, Italy.
Phone: 0039 0912192649
Fax: 0039 0912192422
Email: agallo@ismett.edu
Response to Reviewer 1 Comments
Point 1: This manuscript, in my opinion, can definitely be accepted, but only after certain text edits and supplementations. Despite the fact that the manuscript is written in acceptable English, I would nevertheless recommend editing the entire text as it may significantly improve the text quality.
Response 1: We thank the reviewer for the comment and above all for the useful suggestions. We changed the text accordingly together with the employment of the language service of our Institution by a native English-speaking colleague.
Point 2: Virtually the whole text in the Abstract must be more thoroughly stated; especially, the statement in lines 12-14 should be replaced by "... which causes coronavirus disease 2019 (COVID-19), has altered human activities all over the world and has become a global hazard to public health"; in lines 14-17 " Despite considerable advancements in pandemic containment techniques, in which vaccination played a key role, COVID-19 remains a global threat, particularly for frail patients and unvaccinated individuals who may be more susceptible to developing acute respiratory distress syndrome (ARDS)."; in line 19 "... had a similar survival rate to those with COVID-19-unrelated ARDS."; and in lines 19-25 " In order to shed light on the potential mechanisms underlying the COVID-19 infection, we conducted this proof of concept study using a single-cell V(D)J and gene expression sequencing of the B cell to examine the dynamic changes in transcriptomic BCR repertoire present in patients with COVID-19 at various stages. We compared a recovering and a deceased COVID-19 patient supported by ECMO with one COVID-19-recovered patient who did not receive ECMO treatment and one healthy subject who had never been infected previously.”
Response 2: We thank the reviewer for the suggestions. We changed the text accordingly.
Point 3: In Introduction. Rewrite the sentence in lines 29-30 as follows: " Since December 2019, the severe acute respiratory syndrome coronavirus 2 has posed a hazard to global public health (SARS-CoV-2)."; in lines 33-38, the sentence might be replaced by " The similarities between the worst SARS-CoV-2 consequences and seasonal influenza problems, such as ARDS or multiple organ failure, suggested a role for extracorporeal membrane oxygenation (ECMO) implantation in patients with the most severe pulmonary de-compensation."; in lines 36-38, the sentence might be paraphrased as " It has recently been established that ECMO can be utilized as a rescue therapy due to the temporary replacement of lung and/or cardiac function."; the sentence in lines 40-42 should be replaced by " Clearing SARS-CoV-2 infection and hence influencing patient clinical outcomes is also mediated by humoral and adaptive immune responses."; in lines 55-59, the sentence could be replaced by " In this study, we used a single-cell approach to look at the dynamic changes in transcriptomic BCR repertoire in patients with COVID-19 at various stages, comparing a recovered and a deceased COVID-19 patient supported with ECMO with one COVID-19-recovered patient without ECMO treatment, and one healthy subject never infected before and tested for SARS-CoV-2 antibodies."
Response 3: We thank the reviewer for the suggestions. We changed the text accordingly.
Point 4: In Materials and Methods. In lines 62-65, the sentence should be rewritten as " Four subjects were enrolled in this study at the IRCCS ISMETT: one COVID-19 patient on ECMO (survived), one COVID-19 patient on ECMO (died), one COVID-19 recovered patient without the ECMO condition, and one Healthy control who tested negative for both nasopharyngeal swab (NPS) and anti-Spike and anti-N IgG/IgM."; in lines 68-69, the sentence might be replaced by " Fresh blood samples were acquired from patients on day three of their hospitalization."
Response 4: We thank the reviewer for the suggestions. We changed the text accordingly.
Point 5: In Results. Study Design and Profiling of B Cells. The lines 125-127 should be paraphrased as " The scRNA-seq and single-cell paired BCR analysis were then combined, and the study was limited to 5,830 B lymphocytes having full-length productive paired IGH-IGK/IGL."
In Transcriptional signatures of different B cell subpopulations of COVID-19 patients. Lines 177-180 should be paraphrased as follows " Interestingly, we discovered that MT-ATP8 expression levels were associated to disease severity; in particular, the two COVID-19 patients getting ECMO had near-zero MT-ATP8 expression compared to the no-ECMO patient and the Healthy control."
In Features of B Cell Subsets. Lines 177-185 should be rewritten as "Interestingly, we discovered that MT-ATP8 expression levels were associated to disease severity; in particular, the two COVID-19 patients getting ECMO had near-zero MT-ATP8 expression compared to the no-ECMO patient and the Healthy control. The most notable downregulated gene in activated B cells was ribosomal protein S4, Y-Linked 1 (RPS4Y1), which has been linked to HLA-DQA1 and has been implicated in the resistance to treatment with corticosteroids and cyclosporin A. (Figure 2B)."; in lines 186-188, the sentence might be replaced by " The profiling of plasma cells clusters revealed a considerable downregulation of an Immunoglobulin Heavy Chain Variable Region, IGHV3-73, which has been described to permit antigen binding activity and immunoglobulin receptor binding activity (Figure 2B)."; the sentence in lines203-207, should be replaced by "The distribution of clonally increased B cells in COVID ECMO S, COVID ECMO D, COVID R, and Healthy controls was studied. The percentage of clonally expanded cells in COVID ECMO D patient B cells was 1.97%, which was somewhat higher than in the other three subjects' B cells, specifically: 0.56% in COVID ECMO S, 1.0 206% in COVID R, and 1.14% in the Healthy control (Figure 3C)."; in lines 210-211 " ....for each sample (Figure 3D), with no significant difference between the samples."
Response 5: We thank the reviewer for the suggestions. We changed the text accordingly.
Point 6: In Discussion. In line 245, the first phrase should begin " In the present study, we aimed (or wanted) to..." (not study's goal but the authors' goal); the sentence in lines 248-253 should be written as " The cohort included a COVID-19 patient who died after receiving ECMO (COVID ECMO D), a COVID-19 patient who survived the treatment (COVID ECMO S), a COVID-19 recovered patient without ECMO support (COVID R), and one healthy control who tested negative for both nasopharyngeal swab (NPS) and anti-Spike and anti-N IgG/IgM. "; line 255 should be slightly revised to read "... B cells sequenced, based on the average log fold change of the canonical markers, and we assessed..."
Response 6: We thank the reviewer for the suggestions. We changed the text accordingly.
Point 7: The Discussion (lines 315-330) focuses mainly upon the description of experimental technique rather than the interpretation of the results, and reading the conclusions (lines 315-330) it appears that the objective of this work is the technique for doing it out. So, the question is what is the true meaning of this work?
Response 7: We thank the reviewer for the question. Our aim was firstly to unravel changes in transcriptomic BCR repertoire in patients with COVID-19 at various stages, with a single cell sequencing approach. Unfortunately, due to the number of analyzed subjects, no strong conclusion may rise indeed. For this reason, we can’t state strong conclusion but our preliminary findings are in line with other paper published. Nevertheless, we think, and hope, that the analytical approach used, might result interesting and add knowledge on the field.
Point 8: The statement in lines 332-334 of Limitation of the study (page 10) should be changed by " It is important to highlight that the study's main weakness is the small sample size, which prevents the results from being meaningful for stating any of the processes involved in the immune response to SARS-CoV-2. "; I suggest changing the phrase in lines 334–338 to "However, given the uniqueness of the disease characteristics and the molecular approach used, single-cell sequencing, which is based on a large number of events/cells and analyzes each cell as an independent sample, we hope that our computational analysis will provide novel insights for large population cohort studies.";
Response 8: We thank the reviewer for the suggestions. We changed the text accordingly.
Point 9: I recommend to improve the quality (color sharpness) of Figures 1 D and Figure 2 A.
Response 9: We thank the reviewer for the suggestion. However, the software used for the analysis, Loupe Browser 6.0, does not allow changing colour sharpness of the heat maps generated.
Point 10: Finally, I recommend that the structure/composition or formal scheme of the antigen-binding domain of immunoglobulins be provided not only in the text (as described in lines 46-53), but also illustrated with detailed explanations, so that the reader may better understand the context of this article.
Response 10: We thank the reviewer for the suggestion. As requested we provided a formal scheme of the antigen-binding domain of immunoglobulins as Figure 1.

Reviewer 2 Report
Although authors have performed a significant study of COVID19 patients, it needs to be revised carefully.
Abstracts needs to be rewritten and reorganized, what is missing knowledge and how present study overcame an existing problem.
Provide key findings/observations in the abstract and conclusion sections. Add one or two implications of the study.
Please provide references to methods/procedures sections
Scientific literature in introduction and discussion needs to be enriched. I suggest: Kadam, U.S., Lossie, A.C., Schulz, B., Irudayaraj, J. (2013). Gene Expression Analysis Using Conventional and Imaging Methods. In: Erdmann, V., Barciszewski, J. (eds) DNA and RNA Nanobiotechnologies in Medicine: Diagnosis and Treatment of Diseases. RNA Technologies. Springer, Berlin, Heidelberg. https://doi.org/10.1007/978-3-642-36853-0_6
The Figure’s resolution needs to be improved. Also increase the fonts size in figure legends to make it readable.
Please remove limitation of the study section. Incorporate this information in discussion and conclusions sections. Just adding few sentences (instead of whole para) would be sufficient.
Author Response
Palermo 6th February 2023
Dear Editor,
Please find enclosed a revised version of the Case Report entitled “Proof of concept analysis of B cell receptor repertoire in COVID-19 patients under ECMO by single-cell V(D)J and gene expression sequencing.” ID: cimb-2196562 by Gallo et al.
As required, together with the revised manuscript, we now resubmit a point by point response to the reviewer’s criticisms (see below).
We thank the Referees for their helpful comments that greatly improved the manuscript. We have tried to do our best to respond to the points raised.
The Referees have brought up some good points and we appreciate the opportunity to clarify our research objectives and results. As indicated below, we have checked all the general and specific comments provided by the Referees and have made necessary changes accordingly to their indications.
Sincerely,
Alessia Gallo
Alessia Gallo, PhD
Department of Research,
IRCCS ISMETT, via Tricomi 5, 90127 Palermo, Italy.
Phone: 0039 0912192649
Fax: 0039 0912192422
Email: agallo@ismett.edu
Response to Reviewer 2 Comments
Point 1: Although authors have performed a significant study of COVID19 patients, it needs to be revised carefully.
Abstracts needs to be rewritten and reorganized, what is missing knowledge and how present study overcame an existing problem.
Response 1: We thank the reviewer for the useful comment. As suggested we reorganized and rewrote the abstract.
Point 2: Provide key findings/observations in the abstract and conclusion sections. Add one or two implications of the study.
Response 2: We thank the reviewer for the suggestion. We added in the abstract section the key findings: “Our analysis revealed a downregulation of FXYD, HLA-DRB1 and RPS20 in memory B cells, MTATP8 and HLA-DQA1 in the naïve cells, RPS4Y1 in the activated B cells and IGHV3-73 in the plasma cells of COVID-19 patients. We further described an increased ratio of IgA+IgG to IgD+IgM, suggestive of an intensive memory antibody response, in the COVID ECMO D patient. Finally we assessed a VDJ rearrangement heavy chain toward IgHV3, IGHJ4 and IGHD3/IGHD2 families in COVID-19 patients despite the severity of the disease.” And in the conclusion section: “Our findings uncover that the V(D)J rearrangements of the BCR heavy chain of ECMO COVID-19 patients analyzed are in line with other studies focusing on infected and/or vaccinated against SARS-CoV-2 subjects.”
Point 3: Please provide references to methods/procedures sections
Scientific literature in introduction and discussion needs to be enriched. I suggest: Kadam, U.S., Lossie, A.C., Schulz, B., Irudayaraj, J. (2013). Gene Expression Analysis Using Conventional and Imaging Methods. In: Erdmann, V., Barciszewski, J. (eds) DNA and RNA Nanobiotechnologies in Medicine: Diagnosis and Treatment of Diseases. RNA Technologies. Springer, Berlin, Heidelberg. https://doi.org/10.1007/978-3-642-36853-0_6
Response 3: We thank the reviewer for the suggestion. We added references in the Methods section and the suggested reference in the Discussion section.
Point 4: The Figure’s resolution needs to be improved. Also increase the fonts size in figure legends to make it readable.
Response 4: We thank the reviewer for the suggestion. We increased the font size in figure legends. Regarding the figure’s resolution, we have added high-resolution pictures (600 dpi).
Point 5: Please remove limitation of the study section. Incorporate this information in discussion and conclusions sections. Just adding few sentences (instead of whole para) would be sufficient.
Response 5: We thank the reviewer’s suggestion. We incorporate the limitation of the study paragraph in the conclusion paragraph.

Round 2
Reviewer 2 Report
Authors have revised the contents.